# Dietary Acrylamide: A Detailed Review on Formation, Detection, Mitigation, and Its Health Impacts

**DOI:** 10.3390/foods13040556

**Published:** 2024-02-12

**Authors:** Indira Govindaraju, Maidin Sana, Ishita Chakraborty, Md. Hafizur Rahman, Rajib Biswas, Nirmal Mazumder

**Affiliations:** 1Department of Biophysics, Manipal School of Life Sciences, Manipal Academy of Higher Education, Manipal 576104, Karnataka, India; indira.g@learner.manipal.edu (I.G.); sana.mlscma@learner.manipal.edu (M.S.); ishita.chakraborty@learner.manipal.edu (I.C.); 2Department of Quality Control and Safety Management, Faculty of Food Sciences and Safety, Khulna Agricultural University, Khulna 9100, Bangladesh; 3Department of Physics, Tezpur University, Tezpur 784028, Assam, India; rajib@tezu.ernet.in

**Keywords:** dietary acrylamide, Maillard’s reaction, acrylamide toxicity, E-nose, tongue

## Abstract

In today’s fast-paced world, people increasingly rely on a variety of processed foods due to their busy lifestyles. The enhanced flavors, vibrant colors, and ease of accessibility at reasonable prices have made ready-to-eat foods the easiest and simplest choice to satiate hunger, especially those that undergo thermal processing. However, these foods often contain an unsaturated amide called ‘Acrylamide’, known by its chemical name 2-propenamide, which is a contaminant formed when a carbohydrate- or protein-rich food product is thermally processed at more than 120 °C through methods like frying, baking, or roasting. Consuming foods with elevated levels of acrylamide can induce harmful toxicity such as neurotoxicity, hepatoxicity, cardiovascular toxicity, reproductive toxicity, and prenatal and postnatal toxicity. This review delves into the major pathways and factors influencing acrylamide formation in food, discusses its adverse effects on human health, and explores recent techniques for the detection and mitigation of acrylamide in food. This review could be of interest to a wide audience in the food industry that manufactures processed foods. A multi-faceted strategy is necessary to identify and resolve the factors responsible for the browning of food, ensure safety standards, and preserve essential food quality traits.

## 1. Introduction

For centuries, people have employed thermal processing to prepare food before consumption. The heating process holds several significant benefits for food products, including enhancing nutritional quality, preventing microbial contamination, improving flavor, and extending the shelf life of foods. While thermal processing offers various advantages, it also gives rise to unwanted heat-induced toxins as a by-product, known as “thermally processed contaminants”. Acrylamide is one such chemical that has garnered considerable curiosity in recent years [1].

Acrylamide, known by its chemical name 2-propenamide (C_3_H_5_NO), is an odorless, white, water-soluble, highly reactive, and crystalline solid with a molecular weight of 71.08 gmol^−1^ [2]. It has become a significant concern as it is found in most thermally processed carbohydrate-rich foods. It was categorized as a potential human carcinogen, neurotoxicant, and genotoxicant by the International Agency for Research Cancer (IARC) in 1994. The Swedish National Food Administration (SNFA) was the first to report the presence of acrylamide in foods in 2002 when they were subjected to higher temperatures [3]. Subsequent studies from various countries, including the United Kingdom, the United States of America, Norway, Germany, and Switzerland, confirmed the presence of acrylamide in baked and fried dishes cooked at high temperatures [4]. Potato chips, crisps, coffee, pastry, and sweet biscuits were identified by the Joint Expert Committee on Food Additives (JECFA) as significant sources of acrylamide in daily diets worldwide, particularly in carbohydrate-rich foods cooked at temperatures higher than 120 °C [1].

Since the detection of acrylamide in food in 2002, extensive efforts have been undertaken to address its presence. FoodDrinkEurope’s “Acrylamide toolbox”, last updated in May 2019, serves as a comprehensive guide. A workshop organized by the Commission in January 2014 emphasized sector-specific implementation of mitigation measures, engaging consumer organizations to promote awareness of cooking practices. EFSA’s June 2015 scientific opinion highlighted acrylamide as a carcinogenic substance, expressing concern over inconsistent reductions in dietary exposure. Member States’ investigations revealed varied implementation of mitigation measures, leading to discussions on regulatory actions. The adoption of Commission Regulation (EU) 2017/2158 in November 2017 mandated mitigation measures with benchmark levels for food business operators, and ongoing considerations involve setting maximum acrylamide levels in specific foods. Commission Recommendation (EU) 2019/1888, issued in November 2019, emphasizes monitoring and complements existing regulations, reflecting a comprehensive approach to addressing acrylamide in the European Union.

Dietary carbohydrates play a crucial role in regulating blood glucose levels. Starch is one such essential carbohydrate that is present in the human diet and supplies a consistent energy source. It is a polysaccharide, comprising an extensive network of glucose units that are chemically made up of primary additives amylose, a long-chain polymer with α-1,4-linked linear chains that comprise 20% of the overall starch, and amylopectin, which is a branched-chain polymer with α-1,4-linked glucose units joined by α-1,6-linkages at the branched points, constituting 80% of the overall starch [5]. Among different countries, acrylamide intake varies among individuals based on nutritional habits and food processing [1]. However, adolescents and children consume three times more acrylamide than adults due to their preference for baked and fried starchy foods like French fries, cookies, and potato chips [4]. 

Various factors, such as pH, concentration, food processing parameters like time, temperature, water activity, the presence of free asparagine and reducing sugars, and moisture content, as well as the physical parameter or the surface area ratio of the food product, influence the formation of acrylamide in food [6]. However, there are some mitigation strategies to overcome this problem and reduce the concentration of acrylamide in food, and these include lowering the pH level, the use of additives such as vitamins, amino acids, enzymes, and antioxidants, utilizing blanching soaking processes, hydrocolloid coatings, and controlling the temperature and atmosphere while processing the food. 

Apart from dietary sources, exposure to acrylamide can also occur through alternative means such as inhaling, smoking tobacco, and dermal absorption. Dermal exposure is less critical given that the skin acts as a barrier, reducing the absorption of acrylamide. 

In contrast, exposure to food remains a significant concern [7,8]. In addition, the use of new techniques for the design and development of procedures, such as the use of electronic devices for biosensors, has also been a recent development in determining dietary acrylamide in food and other sources [7]. However, there are very few reviews that extensively provide information about the formation, mitigation, and detection of acrylamide [9,10,11,12,13,14,15]. Therefore, this current updated review primarily focuses on elucidating the mechanism of acrylamide formation in food, exploring factors that influence its occurrence, discussing recent trends in detection and mitigation strategies, and addressing the health risks associated with dietary acrylamide toxicity. Through this comprehensive review, we aim to contribute valuable insights that may aid in a better understanding of processing techniques for carbohydrate-rich thermally processed foods, ultimately aiming to prevent the formation of acrylamide. 

## 2. Materials and Methods

This review focuses on understanding the sources of dietary acrylamide, its potential health effects, and mitigation strategies. We selected the relevant literature using specific databases (PubMed, Scopus, etc.) with the specific keywords (“acrylamide”, “diet”, “health effects”). We considered different types of studies (epidemiological, experimental, and reviews) with specific data points of interest (acrylamide levels, acrylamide exposure, and health outcomes).

Most of the available research articles discuss a particular section regarding dietary acrylamide. However, our review comprehensively discusses the formation of dietary acrylamide, its potential health effects, and mitigation strategies.

## 3. Mechanism of Acrylamide Formation in Food

Initially, acrylamide is absent in the raw materials of foods. However, it is formed when a food ingredient is heated or thermally processed to a temperature up to or above 120 °C through broiling, baking, frying, or roasting [9]. The principal precursors for acrylamide formation are asparagine (C_4_H_8_N_2_O_3_) and reducing sugar. During the heating process of food, free asparagine and the reducing sugars such as glucose or fructose react together, resulting in acrylamide formation; this reaction is known as the “Maillard reaction” [9,10]. It involves a non-enzymatic reaction, a complex series of reactions between the free asparagine present in the food ingredient and the reducing sugars (glucose and fructose) when subjected to an elevated temperature, thereby changing the color and flavor of the food product, which is also known as browning of the food. The aldehyde group of glucose and the keto group of fructose render reducing sugars highly reactive during acrylamide formation [11]. The general reaction for acrylamide formation from the asparagine precursor is illustrated in Figure 1a.

The first step of the “Maillard reaction” involves the condensation of the asparagine with the reactive carbonyl moiety, leading to the production of Schiff bases [12]. These unstable Schiff bases undergo Amadori rearrangement at an acidic pH, resulting in the Amadori compound [11,13]. Subsequently, the Amadori compound undergoes enolization to form reductones and deoxyosones, ultimately converting to melanoidin compounds that impart a brown color and aroma to the food. The dicarbonyl compounds undergo decarboxylation to produce Strecker aldehyde, which will favor acrylamide formation in food; this mechanism is known as “Strecker degradation” [14]. Another pathway involves the decarboxylation process of the Schiff base to Azomethine ylide [12]. The decarboxylated product undergoes hydrolyzation, forming 3-Aminopropionamide, which further degrades to form acrylamide [15]. The overall mechanism of acrylamide formation via the Maillard reaction is represented in Figure 1a.

Beyond the Maillard reaction, several minor pathways contribute to acrylamide formation in food; one such pathway is the “Acrolein pathway”. Acrolein, known by its chemical name 2-propenal, is a compound of great concern for its carcinogenicity [16]. Acrolein can be formed through the thermolysis or decomposition of glycerols at high temperatures, also called pyrolysis of glycerols [17]. Acrolein then undergoes oxidation to produce acrylic acid, which, when reacting with ammonia produced during Strecker degradation, forms acrylamide [18]. Acrylic acid can also be produced from aspartic acid due to the Maillard reaction and acrylamide formation from asparagine [19]. Typically, when fats or oils, particularly unsaturated oils, are subjected to considerably high temperatures until they emit an unpleasant smoke, as glycerol undergoes dehydration, this leads to the formation of acrolein and. consequently, the production of acrylamide.

Acrylamide can not only be formed from carbohydrates via the Maillard reaction, but it can also be formed from other macromolecules such as lipids through the acrolein pathway, proteinogenic amino acids like serine and cysteine through the pyruvic acid pathway, and from aspartic acids, β-alanine and carnosine, which are all capable of producing acrylamide [20]. Acrylamide has also been found to form in meat products via the acrolein pathway when subjected to higher temperatures, resulting from lipid degradation leading to the production of acrolein, which then oxidizes to form the acrylamide content in meat samples, as reported by Lee et al. [21].

## 4. Factors Influencing the Formation of AA 

The formation of acrylamide in food depends on several factors during the cooking process, such as the cooking temperature, time, pH, surface/area ratio, moisture content, cooking method, use of fertilizer, harvesting, and storage conditions. These factors are discussed in detail.

### 4.1. Cooking Temperature and Time

Previous studies have shown that the acrylamide content in food products exponentially increases with higher temperatures. At higher temperatures, the amino group of the free asparagine present in the food undergoes condensation with the carbonyl group of the reducing sugars, forming an intermediate compound, Schiff base, which is responsible for the formation of acrylamide [22]. However, prolonged heating shows decreased acrylamide content due to degradation [23]. Whereas cooking food products at a lower temperature for a prolonged time reduces the acrylamide content [24]. 

### 4.2. pH and Surface Area

The formation of acrylamide in food diminishes under lower pH conditions during the frying or baking of carbohydrate-rich foods [20]. In the mechanism of acrylamide formation through the Maillard reaction, the processes involving the formation of Schiff base, Heyns rearrangement, and the breakdown or formation of Amadori compounds are predominantly influenced by pH. The conversion of the non-protonated amine group to protonated amine at lower pH obstructs the reaction of acrylamide formation, consequently reducing the acrylamide content in food [25]. Whereas alkaline conditions promote the formation of acrylamide content in food. Under alkaline conditions, the free amino group undergoes deprotonation, forming nucleophile molecules that can readily attack the carbonyl moiety. This initiates the formation of the intermediate compound glycosylamine, thus triggering the formation of acrylamide via the Maillard reaction [26]. 

The slice thickness of the food product also affects acrylamide formation; thin pieces or slices reduce the acrylamide content due to a higher surface area ratio. In contrast, thick slice pieces would favor the formation of acrylamide, as it requires a higher temperature to reach the core during cooking. This is evident in a study where potato slices cut at a thickness of 1/4’’ exhibited a linear increase in the acrylamide content, reaching 3402 µg kg^−1^, compared to slices of 1/2’’ thickness [27]. 

### 4.3. Moisture Content

Moisture content refers to the amount of unbound water in food products, which plays a significant role in the formation of acrylamide in food products. The moisture content of the food decreases with an increase in temperature, promoting acrylamide formation [28]. In the case of bread, the moisture content alters the acrylamide concentration and the nature of the dough during the baking process. A lower moisture content is likely to be associated with the Maillard reaction, which plays a significant role in the decarboxylation of the Amadori compound. This, in turn, favors the formation of acrylamide in the food by not hindering the reaction caused by excess water molecules, as reported by [29]. Considering that the moisture content at the external surface of food products is much lower than the internal surface, acrylamide formation tends to be higher at the outer surface of food products such as French fries, cookies, chicken nuggets, sweet biscuits, and breakfast cereals, as reported by [30]. In addition to moisture content, water activity is also associated in influencing acrylamide formation; when the water activity ranges between 0.4 and 0.8 and the moisture content is <5% it is likely to favor the Maillard reaction, subsequently leading to the formation acrylamide in food products [1].

### 4.4. Food Composition

The formation of acrylamide depends on the food composition. Various studies have reported that the presence of exogenous phenolics enhances the formation of acrylamide. However, the phenolic antioxidants did not affect the acrylamide formation [15]. Further, the presence of free amino acids (glycine, alanine, lysine, glutamine, and glutamic acid) and proteins was shown to reduce the acrylamide content in potatoes [29,30,31]. Whereas the presence of free carbohydrates such as glucose and fructose increased the formation of acrylamide by 3- and 8-fold and the fructose level by 14- and 50-fold.

### 4.5. Cooking Method 

Acrylamide formation is comparatively lower when food is cooked using a moisture-less oven or dry heating than in deep-frying, where the degree or level of browning is high. In contrast, the degree of browning in a moisture-less oven is slight, and acrylamide formation was observed at temperatures above 120 °C. However, in some cases, microwave cooking influences the acrylamide production as it favors the condition for the Maillard reaction compared to traditional cooking methods [22]. Cooking meat, chicken, and shrimp nuggets using the traditional cooking method increased the acrylamide content compared to the industrial cooking method, owing to differences in the cooking temperature and time. The lowest acrylamide content was found in chicken nuggets at 7.3 ng g^−1^ fried using the industrial process for 3 min at 180 °C. The highest was found in shrimp nuggets at 27 ng g^−1^, which were fried using the traditional method for 6 min at 220 °C, as reported by [31].

### 4.6. Use of Fertilizer, Harvesting, and Storage Condition

Since the use of fertilizer is considered an essential factor in crop production, several studies have reported the effect of fertilization on an increase in the acrylamide content. Limited use of fertilizers, such as nitrogen, increases the acrylamide content by elevating the concentration of reducing sugars in potato tubers. Harvesting also plays a crucial role in affecting the acrylamide content of food products, with the acrylamide content tending to decrease as the growing time increases. It also depends on the climatic conditions; cold weather ranging from 8 to 12 °C and warm weather ranging from 25 to 30 °C tend to increase the acrylamide content in the food product [1]. The concentration of acrylamide and glucose increased as storage time increased due to senescent sweetening. There is more accumulation of the reducing sugar molecules, an irreversible process observed in potato tubers [32]. The same author correlated the fertilization level and storage time in another study and observed a linear increase in acrylamide and asparagine content with an increase in storage time and N-fertilization [33]. Thus, storing cooked food at a lower temperature for an extended period of time can increase the formation rate of acrylamide [34]. 

All of the above-discussed factors influence acrylamide formation in food, leading to several harmful health effects in humans. Cooking at very high temperatures, lower moisture content, and alkaline conditions can rapidly increase the formation of acrylamide content in the food. Therefore, understanding the mechanism and interaction of acrylamide with biological molecules is an essential and challenging topic. 

## 5. Diseases Related to Acrylamide Exposure

Dietary acrylamide, due to its high toxicity, can cause harmful health effects such as genotoxicity, neurotoxicity, hepatoxicity, carcinogenicity, cardiovascular toxicity, reproductive toxicity, and prenatal–postnatal developmental toxicity. All of these toxic effects of acrylamide have been comprehensively evaluated and categorized by the Joint Expert Committee on Food Additives (JECFA) and the International Agency for Research on Cancer (IARC) [9].

### 5.1. Genotoxicity and Carcinogenicity

The presence of acrylamide in food has been identified as mutagenic. Several studies have indicated its ability to induce sister chromatid exchanges, resulting in chromosomal aberrations like aneuploidy, polyploidy, and several other mitogenic disturbances [35]. In humans, acrylamide has a pronounced impact on cells, forming adducts with nucleic acid bases that lead to the production of some undesired components, potentially causing genotoxic effects. Several new studies have reported positive results on the clastogenic and mutagenic properties of acrylamide (AA) and its active metabolitel glycidamide (GA). Glycidamide is a genotoxic component of acrylamide that is formed by the oxidation process of acrylamide involving cytochrome. Acrylamide itself is an inactive mutagen and lacks metabolic activity; therefore, it exhibits limited reactivity towards DNA. However, upon biotransformation into glycidamide, it becomes capable of inducing mutations, leading to genotoxic effects. DNA adducts of GA were induced by AA exposure in experimental animals and have also been observed in humans. This genotoxic glycidamide significantly affects the lymphocytes upon reaching the bloodstream, causing lesions in lymphocytic DNA, as evidenced by a comet assay analysis. Biotransformation of AA into genotoxic glycidamide (GA) in rat hepatocytes occurs at a slower rate than detoxification via glutathione (GS), indicating complexities in its metabolic pathways. Current dietary exposure estimates suggest that the margins of exposure (MOEs) for acrylamide (AA) are below 500. However, arguments propose that AA may not function as a genotoxic carcinogen, particularly at levels relevant to consumers [36]. However, the European Food Safety Authority (EFSA) has examined these recent studies and determined that the margin of exposure (MOE) method remains suitable for assessing the risk associated with the carcinogenic effects of AA. [37]. Acrylamide, along with its metabolite glycidamide, has been implicated in inducing mutations in the tumor suppressor gene TP53. Human TP53 knock-in fibroblasts of the mouse embryo subjected to acrylamide exposure showed mutations that led to tumor development in the ovary, lungs, and breast, as reported by Hölzl-Armstrong et al. [38]. In another study utilizing a mouse model, acrylamide induced genotoxicity in bone marrow when orally administered at varying concentrations ranging from 0 to 32 mg kg^−1^ for 30 days. This resulted in a significant increase in chromosome aberrations and a corresponding decrease in the index in the bone marrow cells, as reported by Algarni [39]. The visual representation of the genotoxic effect of dietary acrylamide on humans is shown in Figure 2a.

Acrylamide has been classified based on experimental research as a “probable human carcinogen” by the International Agency for Research on Cancer (IARC) in 1994 and a “reasonably anticipated human carcinogen” by the US National Toxicology and Reports on carcinogen in 2011. Experimental research, including studies cited, supports acrylamide’s carcinogenicity in humans, attributed to its formation of adducts with DNA and other biomolecules like hemoglobin. This induces gene mutations and chromosomal abnormalities that would cause cancer [40]. According to the studies, the genotoxic mechanism can be a pathway for the carcinogenic effect of acrylamide in humans, where the acrylamide is biotransformed into its metabolite glycidamide involving cytochrome. These induce chromosomal aberrations, sister chromatid exchanges, and DNA fragmentation, leading to carcinogenic effects ultimately causing cancer [41]. Both acrylamide and its metabolite will covalently bind to the nucleophilic site of the biomolecules, particularly DNA, proteins, and hemoglobin molecules, forming adducts and inducing lung cancer, as observed in mouse embryonic fibroblasts. When they develop an adduct with the hemoglobin molecules, it is used as a biomarker for acrylamide and glycidamide exposures [42]. 

Some long-term studies have reported that acrylamide exposure in humans can cause cancer in the ovaries, prostate gland, lungs, brain, reproductive tract, thyroid, and intestine, which exhibit hemoglobin DNA adducts, as shown in Figure 2b [43]. The intake of acrylamide-rich food will increase malignant and benign tumors in organs like the thyroid, reproductive organs, and adrenals, which would lead to hormonal imbalance.

A study observed an acrylamide-associated cancer risk in the endoderm and women’s ovaries with at least one copy of the GSTT1 gene. Increased risk was also associated with a Single Nucleotide Polymorphism (SNP) in the CYP11A1 gene responsible for the synthesis of progesterone hormone [44]. However, conflicting human studies have reported no association between dietary acrylamide and ovarian cancer risk beyond factors like smoking and alcohol consumption [45]. 

Acrylamide exposure induced oxidative stress and an increase in malondialdehyde (MDA) levels, leading to lipid peroxidation and histopathological changes involving degeneration of the focal gliosis in the brain. This was observed in a rat model that was administered acrylamide orally with concentrations ranging from 5 to 20 mg kg^−1^ day^−1^ for 60 days [46]. Acrylamide exposure increased Tumor Necrosis Factor–Alpha (TNF-α) levels, causing inflammation in the brain tissues by the release of various cytokines due to oxidative stress induced by acrylamide [47]. Acrylamide exposure can cause polymorphisms in the gene responsible for the functioning of the cytochrome in the lungs, thereby affecting the metabolism of acrylamide in the human body, which was analyzed using the dried blood spot samples of lung cancer patients [48]. In other studies, exposure to acrylamide can induce the formation of DNA adducts in the lungs, leading to histone modification and epigenetic changes when examined using mice model administered with 0.70 mM of acrylamide in drinking water [49]. Acrylamide exposure decreased the Thyroid Stimulating Hormone (TSH) level, or serum T4, leading to increased proliferation and tumorigenesis when female rats were exposed to 50 mg/kg of acrylamide in drinking water [50]. Elevated acrylamide exposure may result in oxidative stress, creating a deficit in antioxidants by diminishing catalase activity. This reduction in catalase function, crucial for shielding against oxidative stress, is associated with an increased risk of prostate cancer in human males aged 55–69 years, according to studies conducted on this demographic group [51].

### 5.2. Neurotoxicity

The European Food Safety Authority (EFSA) summarized experimental data in 2015 and reported that acrylamide can induce adverse neurotoxic effects in humans through occupational skin absorption and peripheral neuropathy [8]. As a well-known neurotoxin, acrylamide affects both the central and peripheral nervous systems, manifesting in general symptoms such as weakness of the skeletal muscles, axon degradation in the peripheral and central nervous systems, distal edema, loss of weight, and ataxia [1]. The brain is particularly susceptible to acrylamide’s effects as they contain double bonds and are highly reactive to thiols, leading to peripheral neuropathy, characterized by weakness or pain in the hands and feet due to nerve damage. This condition is mainly caused by oxidative stress induced by free radicals and Reactive Oxygen Species (ROS) in the sciatic nerve. Acrylamide exerts significant effects on the sciatic nerve, causing demyelination and degeneration of the nerve fibers. Also, it alters the structure of the myelin membrane, causing deterioration of the axon as examined using a rat’s brain model with an acrylamide intoxication of 10 mg/kg/day as reported by Farag et al. [52]. Acrylamide increases malondialdehyde levels in the spinal cord and brain while decreasing glutathione levels, leading to abnormal gait. It also caused glial cell activation and necrosis in the sciatic nerve, cerebellum, cerebrum, spinal cord, and hippocampus when examined using a rat model treated with 50 mg kg^−1^ bwt day^−1^ [53]. Elevated Reactive Oxygen Species (ROS), 8-hydroxy-2-deoxyguanosine, and malondialdehyde content cause oxidative imbalance, reducing glutathione (GSH) content in mice administered with 5–50 mg kg^−1^ of acrylamide content as reported by Zhao et al. [54]. In some studies, the neurotoxic effect of acrylamide was also observed in the central nervous system, where an increase in pyknotic neuronal cells was observed, potentially inducing apoptotic cell death of granular and Purkinje cells and alterations in the cerebral cortex [55]. Besides oxidative stress, another possible mechanism of acrylamide leading to neurotoxicity is its conjugation with the cysteine residues of proteins present in the presynaptic membrane, disrupting vesicular and presynaptic membrane interactions, leading to the degeneration of the nerve terminal and disrupted neurotransmitter release [56]. Acrylamide affects the small intestine as it is the major absorption route and damages the duodenal Enteric Nervous System (ENS) neurons. An experiment was performed using a pig model, administered with lower doses of 0.5 µg kg^−1^ b.w day^−1^ and higher doses of 5 µg kg^−1^ b.w day^−1^ for four weeks. It was observed that even low amounts of acrylamide altered the ENS neurons, which can induce gastrointestinal disorders [57]. Redox imbalance in the brain resulting from increased malondialdehyde production can cause damage to the neuronal cells, causing neurodegeneration as lipid peroxidation increases in the blood, erythrocytes, and neurofibrillary tangles in the brain, potentially contributing to Alzheimer’s disease [36]. Acrylamide administration at a concentration of 10 mg kg^−1^day^−1^ to a rat model for seven weeks induced memory alterations, disrupted spatial learning, and caused hippocampus neuron loss due to glial cell activation and the accumulation of hyperphosphorylated tau proteins in the cortex and hippocampus, as reported by Yan et al. [58]. The comprehensive neurotoxic effects of dietary acrylamide in humans are shown in Figure 3. 

### 5.3. Hepatoxicity

The liver, being the initial site for dietary acrylamide metabolism, is particularly vulnerable to acrylamide toxicity. One of the most common mechanisms is the formation of oxidative stress in the liver, which generates ROS that intensifies lipid peroxidation, causing oxidative damage in the liver [59]. Acrylamide triggers lipid peroxidation by generating free radicals due to oxidative imbalance. Reactive fatty acids interact with ROS to form lipid hydroperoxides and peroxyl radicals, ultimately producing malondialdehyde (MDA) and significantly increasing lipid peroxidation, resulting in elevated oxidative stress in the liver, as reported by Al-Qahtani et al. [60]. Exposure to acrylamide led to a deficiency in glutathione (GSH) as well as glutathione-S-transferase (GST) activity in the liver tissues, attributed to increased ROS content, thereby damaging the hepatic cells. This was observed in 8–12-week-old female rats fed with 500 µg kg^−1^ day^−1^ of acrylamide through tap water [61]. An increase in oxidative stress can cause DNA damage and disrupt the cell metabolism, causing cell death, leading to hepatoxicity and causing hepatic diseases [62]. An increase in oxidative stress and a reduction in GSH determined the presence of hepatic fibrosis and collagen fiber in the hepatic cells, which reduces the blood flow in the liver, as observed using a male albino rat model that was administered with 50 mg day^−1^ of acrylamide for 14 days [62]. Oxidation of biomolecules, such as proteins, lipids, and DNA, resulting from oxidative stress, can damage cell metabolism, leading to undesired cell death and DNA fragmentation, which would decrease the enzyme rate and cause liver damage. Some studies have revealed that hepatocytes undergo degeneration, forming large lipid vacuoles due to mitochondrial swelling and fatty lipid deposition. This results in vacuolization, congested central veins, and chromatolysis, causing hypoxia, lipid accumulation in the cells, and ischemia. These alterations were observed in male Wistar rats with acrylamide administration of 250 mg kg^−1^ day^−1^ for 21 days [63]. These modifications showed infiltration of mononuclear inflammatory cells, hypertrophied Kupffer cells, necrosis of hepatocytes, hepatic sinusoids, and congested liver blood vessels, mainly in the portal area of the liver. As a result, the serum urea level in the liver was significantly reduced, reflecting impaired urea synthesis, as urea is synthesized in the liver from dietary proteins. This was demonstrated in female Albino Wister rats administered acrylamide concentrations ranging from 2 to 30 mg kg^−1^ of body weight per day.

The hepatotoxic effect of acrylamide in the liver also extends to lipid metabolism. Given the central role of the liver in lipid metabolism, oxidative stress induced by acrylamide can pathologically or physiologically impact lipid metabolism, causing damage to liver tissues [64]. The factors influencing the hepatoxic effect in the liver and the causes associated with it are represented in Figure 4.

### 5.4. Cardiovascular Toxicity

Acrylamide acts as a xenobiotic in the blood, inducing oxidative stress in the erythrocytes through lipid peroxidation. A study was conducted where a zebrafish model was administered acrylamide with concentrations ranging from 0.5 to 5 mmol L^−1^ observed alterations in redox signaling affecting the physiological development of the cardiac system. Acrylamide causes heart dysfunction and malformation, decreasing the number of cardiomyocytes and reducing their proliferation capacity. Moreover, it was observed that acrylamide alters the atrioventricular canal differentiation using whole-mount in-situ hybridization using various biomarkers, as reported by Huang et al. [65]. Another study by the same author, employing the zebrafish embryo model, highlighted cardiac developmental toxicity caused by acrylamide. The observed effects included a shrunken heart with impaired function and morphogenesis. It also decreased the number of endocardial and myocardial cells, which altered the development of the atrium and ventricle. They also observed a reduction in gene expression during cardiogenesis. Acrylamide exposure during the first trimester of human pregnancy was found to induce deficiencies and alterations in initial heart development [66]. The overall representation of the cardiovascular toxicity caused by acrylamide is shown in Figure 5.

### 5.5. Reproductive Toxicity and Prenatal and Postnatal Effects of Acrylamide

Reproductive toxicity caused by acrylamide is a significant concern, as it can lead to infertility. Reproductive toxicity usually affects males in the reproductive organs, such as the epididymis, testis, seminal vesicles, and prostate gland, and causes severe toxicological damage. This toxic substance causes abnormalities in sperm morphology, subsequently reducing motility and sperm count. A study conducted on 30-day-old male albino rats exposed to 10 mg kg^−1^ day^−1^ of acrylamide for eight weeks demonstrated these reproductive effects [67]. 

Leydig cells secrete testosterone, which plays a vital role in the reproductive development of males. As reported in a study, exposure to acrylamide and its toxic metabolites can inhibit progesterone hormone production, causing apoptosis of the Leydig cells as a decrease in the steroidogenic acute regulatory (StAR) protein expression when observed using male rat Leydig cells [68]. Acrylamide generates Reactive Oxygen Species (ROS) in Leydig cells, causing DNA damage and lipid peroxidation, resulting in oxidative stress leading to cell death. Acrylamide and its metabolite glycidamide generate excessive ROS in the mitochondria, affecting protein transport through membranes and causing membrane depolarization, ultimately impacting steroid hormone synthesis in Leydig cells, as reported by Sun et al. [32]. Some studies indicate acrylamide-induced hyperplasia of Leydig cells, which can influence spermatogenesis and limit testosterone circulation, potentially impairing male reproductive function [69]. Additionally, acrylamide has been linked to decreased serum testosterone levels, reducing Leydig cell size and count, thereby compromising responsiveness to the gonadotrophic signal [70].

In females, acrylamide targets the ovary, impacting corpus luteum formation, atresia, and follicular development. Nitric Oxide Synthase (NOS) involvement during follicular oocyte formation, development, and atresia leads to alterations in corpus luteum size and primordial follicles, and an increase in the proportion of the primary follicles was observed in the study reported by Wei et al. [71]. The quality of oocytes is crucial for maturation and development, and dietary acrylamide was found to significantly affect the quality of the egg cells. A study reported that a notable percentage of the oocytes could not enter the meiosis process, caused by the disruption of spindle organization due to damage in the centrosome proteins, hindering oocyte maturation and development. Human embryos and oocytes are highly reactive to Reactive Oxygen Species (ROS), leading to damage and resulting in growth and developmental failure, as reported by Duan et al. [72]. The overall representation of the reproductive toxicity of acrylamide is illustrated in Figure 6b.

Several exciting studies have explored the impact of dietary acrylamide on both maternal and paternal health. Consuming dietary acrylamide has been associated with the umbilical cord estradiol levels during delivery, altering the sex hormones during pregnancy [73]. Some studies have reported that acrylamide induces kidney tissue damage in pregnant women, leading to cell inflammation, degeneration of tubular cells leading to massive necrosis of cells, vascular congestion, and Bowman’s [74].

Dietary acrylamide significantly influences the prenatal and postnatal development of infants. Pregnant women consuming acrylamide may directly affect offspring, as acrylamide and its metabolite glycidamide can pass through placental barriers, targeting and disrupting fetal tissues or organs during gestation [75]. They are water-soluble components and thus can target and disrupt the fetal tissues or organs during the gestation period. The postnatal effect of acrylamide on offspring is through breast milk, where it receives poor lactation due to the impact on maternal behavior due to prolactin reduction in the mammary glands, leading to malnutrition and weight loss in the newborn. Intake of foods rich in dietary acrylamide has been associated with an increased risk of small gestational age (SGA), affecting fetal growth in terms of length and weight, as observed in experimental and epidemiological studies [76]. Acrylamide exposure can also lead to lower cord blood Thyroid Stimulating Hormone (TSH), leading to reduced insulin levels in the cord blood [77].

Based on the experimental studies, the leading cause of acrylamide in prenatal and postnatal development is oxidative imbalance, causing oxidative stress [74]. An increase in lipid peroxidation due to a reduction in glutathione (GSH) will generate free radicals, causing oxidative stress during the prenatal stage of development and disrupting placental structure [78]. During pregnancy, the intake of acrylamide-rich foods can increase oxidative stress with a higher risk of maternal inflammation [79]. Oxidative stress, in turn, can cause prenatal and postnatal neurotoxicity by causing morphological changes in the developing cerebellum, reducing the antioxidative defense system [8].

Oxidative stress is implicated in damaging kidney tissues during both prenatal and postnatal development, as the kidney has a reduced antioxidant defense system as compared to other organs; as a result, there is toxic accumulation in the tubules, as reported by Erdemli et al. [74]. Furthermore, elevated oxidative stress can impair bone development, leading to bone loss and altering bone cell differentiation [80]. The overall representation of the prenatal and postnatal effects of acrylamide is shown in Figure 6b. Acrylamide and its metabolite glycidamide induce harmful toxicological effects by generating ROS, leading to oxidative stress in various human organs, resulting in apoptotic cell death, abnormal cell proliferation, and inhibition of infant pre- and postnatal development. 

## 6. Detection of Acrylamide in Food

As acrylamide is found in food products and poses a significant threat to human health, determining its content has become an intriguing challenge. Many experimental studies have been conducted to assess the dietary acrylamide content in food products. There are several detection methods for the determination of acrylamide content in food products, such as electronic tongue and electronic nose, Surface Enhanced Raman Spectroscopy (SERS), Gas Chromatography–Mass Spectroscopy (GC–MS), Liquid Chromatography –tandem Mass Spectroscopy (LC–MS/MS), hemoglobin nanoparticles (HbNPs), and fluorescent biosensors.

### 6.1. Electronic Tongue and Nose

An electronic tongue, or E-tongue, is a potentiometric sensor device capable of detecting the acrylamide content in food through electrostatic interactions [81]. It comprises polymeric lipid sensor membranes that interact with the acrylamide’s amine group, generating potentiometric signals (Figure 7). It was examined in a study using the liquid state of acrylamide solutions in a homemade two-cylindrical sensor array with lipid polymeric membranes. The samples were prepared by homogenizing the fine-grained olive to aqueous paste using deionized water, and the brine solutions were also diluted using deionized water to obtain the same proportions as olive. E-tongue membranes showed an increased response with a rising acrylamide content in olive oil and brine solutions, as reported by Martin-Vertedor et al. [82]. In contrast, an electronic nose, or E-nose, is a sensor device with four gas sensor chips and metal oxide sensors and a data collection block that evaluates organic compounds responsible for the odor of cooked food. The microdetector collects values detected by the sensors, determining the food’s acrylamide content. Thermal treatment can change the aromatic and phenolic profiles of olive oil, thereby altering the acrylamide content of the food. A study established a linear relationship between the aromatic status of the olive oil and the acrylamide content using E-nose, where the samples were prepared by crushing the olive, homogenizing by using Mili-Q water, and centrifugating at 4 °C [83]. These electronic devices surpass conventional methods as they are cost-effective, rapid, user-friendly, efficient, reliable, and accurate for detecting acrylamide content in an aqueous state; see Figure 7.

### 6.2. Liquid Chromatography–Tandem Mass Spectrometry (LC–MS/MS)

Liquid Chromatography–Mass Spectrometry (LC–MS/MS) is one of the most precise and accurate methods for detecting the acrylamide content in food. Liquid Chromatography is an analytical technique used to determine the water-soluble compounds that are non-volatile. As roasted coffee contains toxic components like acrylamide, several experiments were conducted to determine the specific method for the detection of acrylamide content in coffee. In a study report, the acrylamide content of the coffee was determined by using LC-MS/MS with different brands of Turkish coffee. The highest acrylamide content was observed in instant coffee, ranging from 16.5 to 79.5 ng mL^−1^ [84]. Whereas Mesías et al. [85] detected the acrylamide content in the breakfast cereals, including fruits and nuts, using the same above-mentioned method. The acrylamide content was highest in wheat products, ranging from 197 to 639 µg/kg when baked for 35 min at 200 °C. The authors of [86] analyzed and observed the fact that processed food contained a higher amount of acrylamide content and have quoted that this is the reason for the increase in the acrylamide content in the food. The acrylamide content of the medicine homologous food *Atractylodis Macrocephalae* Rhizoma was detected by the above method using graphite carbon nanotubes as an extraction sorbet under processing temperatures ranging from 80 to 210 °C over 5 min to 2 h. The acrylamide concentration was highest with a concentration of 9826 µg/kg when processed at 150 °C for 60 min, as reported by Zhu et al. [87].

### 6.3. Gas Chromatography–Mass Spectrometry (GC–MS)

Gas Chromatography–Mass Spectrometry (GC–MS) is another method for detecting volatile acrylamide in food products. [88]. An experimental study used GC-MS coupled with microextraction and derivatization to determine the acrylamide content in cereal food products. The limit of detection was observed to be 0.6 ng g^−1,^ and the acrylamide content of the samples, such as wafers, was determined to be more than 100 ng g^−1^. Baking temperature and time also influenced the acrylamide level of food products when determined using the above method, as reported by Nematollahi et al. [89]. Moreover, in another study, roasted seeds and nuts were examined by dispersive liquid–liquid microextraction followed by GC-MS, and acrylamide levels ranged between 33.36 and 250.90 µg kg^−1^ due to variations in the roasting temperature and time as observed by Nematollahi et al. [90].

### 6.4. Hemoglobin Nanoparticles (HbNPs)

Proteins such as hemoglobin are redox-active and are used as biosensors in detecting acrylamide content in food products as they have a configuration similar to that of glycidamide. These can induce the formation of hemoglobin adducts, causing harmful health effects. Therefore, HbNPs were prepared and used to detect the acrylamide content in foods such as bread, nuts, potato crisps, biscuits, and also snacks like Kurkure by optimizing the pH, time, and acrylamide concentration and were evaluated by various parameters such as the limit of detection (LOD) showing 0.1 nmol L^−1^, linearity, and recovery percentage as reported in a study by Yadav et al. [91]. In another study, hemoglobin was fixed with the carbon paste electrode to determine the acrylamide content, especially in French fries, which showed the limit of detection of 0.06 nmoL L^−1^ [92]. However, the hemoglobin carbon ionic liquid paste electrode showed a low limit of detection of 5 × 10^−3^ nmol L^−1^ M, as reported in a study by Li et al. [93].

### 6.5. Fluorescence Biosensor

Due to the high sensitivity and selectivity properties of the fluorescent, it has been of great significance in the field of the food industry in determining the acrylamide content. Fluorescence biosensors, utilizing gold nanoparticles, double-stranded DNA, and carbon quantum dots (CQDs), demonstrate high sensitivity in detecting acrylamide content in food products. This method produces a difference in the fluorescence intensity based on the absence or presence of acrylamide compounds in the food. In the presence of acrylamide, adducts and single-stranded DNA are formed. Therefore, the complementary strand gets absorbed on the surface of the gold nanoparticle, thereby quenching, as reported in a study by Asnaashari et al. [94]. Moreover, fluorescent biosensors based on carbon quantum dots (CQDs) and single-stranded DNA, in the presence of acrylamide, these single-stranded DNA were bound to acrylamide through hydrogen bonding. The presence of acrylamide content in the food showed a higher fluorescence signal than without acrylamide, as reported in other studies [95]. 

### 6.6. Surface-Enhanced Raman Spectroscopy (SERS)

Since the 1970s, the intensity of the Raman signal has attained great significance in various fields. Acrylamide content in the fried food was detected using SERS, synthesizing re-oxidized graphene oxide (rGO)/Au nanoparticles. This synthesized rGO/Au composite was mixed with the food sample and observed using a Raman spectrometer at the peak of △v = 1478 cm^−1^ [96]. However, using SiO_2_/Ag nanocomposite immersed in filter paper to detect the acrylamide content using the same above method in food products such as cookies, bread, and chips showed no or weak detection signaling with the limit of detection of 0.02 nmol L^−1^, as reported by Wu et al. [97]. Biodegradable gold-based SERS detected the presence of acrylamide content in the food by showing the peak at 1447 cm^−1^ in an aqueous solution.

Several detection methods as described above showed a great detection range, among which *E-tongue* and *E-nose* are quick and cost-effective, user-friendly, and accurate with an LOD of 2.5 ng g^−1^ compared to other conventional methods, whereas HbNPs are much more specific and sensitive in detecting the acrylamide content in food due to their catalytical activity with an LOD of 0.06 nmol L^−1^ as shown in Table 1.

## 7. Mitigation Strategies of Dietary Acrylamide

Several studies have been conducted to reduce the amount of acrylamide content in carbohydrate-rich foods at the industrial and household levels. The Food and Drug Administration (FDA) and European Regulation (EU) reported mitigation strategies to reduce the acrylamide content in food. It was a challenge to the food industry to reduce the acrylamide content of the food without changing the texture, taste, or appearance of the food product. There are several mitigation strategies to overcome the effect of acrylamide content in food, such as air and vacuum-frying, blanching, additives, pH and water content, fermentation, hydrocolloid coating, and cooking under a controlled atmosphere.

### 7.1. Air and Vacuum-Frying

The air-frying strategy has proven effective in reducing the acrylamide content of food. Vacuum-frying is another method of frying to avoid acrylamide formation conducted at a lower temperature and minimum pressure, which is below 6.65 kPa. It consists of a vacuum chamber, a pump to provide low pressure, and a refrigerator condenser to collect the rising steam [98]. Vacuum-frying of the potato crisps reduced the acrylamide content in the food even among the potato containing a higher level of reducing sugar molecules, not exceeding 250 µg kg^−1^ and moisture content <2.5%, as vacuum-frying operates under lower temperatures, limiting unwanted oxidation processes [99]. Vacuum-frying also significantly reduces the acrylamide content of the food. It removes water content by maintaining a lower temperature under pressure, thereby reducing the concentration of Maillard reaction precursors. Moreover, in other reports, the vacuum-frying method reduced the acrylamide content of the food products, ranging from 72 to 98%, with a moisture content of 4.83% when compared with the conventional baking method at the same moisture level [100].

### 7.2. Blanching

Blanching, a process where food products are soaked in hot water for some time before cooking, has proven effective in reducing the acrylamide content in food. This process enhances texture, maintains uniformity [15], and prevents the enzymatic browning of food by eliminating the soluble sugar molecules [98]. The blanching process leaches out the glucose and asparagine contents. It inactivates the enzymes, which reduces the precursors required for the Maillard reaction, thereby reducing the acrylamide content in the food. In an experimental study, the blanching process decreased the acrylamide content in the food till the end of the gastric stage of digestion, which can be correlated to the decrease in Schiff base as the free asparagine and reducing sugar contents are leached out [30]. Blanching temperature and time also play a vital role in reducing the acrylamide content. Blanching at a higher temperature, about 70 °C, for a shorter period (10–15 min), was much more efficient as it lowered the acrylamide content [27].

### 7.3. Addition of Additives

The addition of substances like amino acids, antioxidants, enzymes, salts, and vitamins can reduce the acrylamide content in food [1]. The addition of amino acids such as lysine, glycine, and alanine reduced the acrylamide content in the food as it has a nucleophilic component that binds covalently, thus eliminating acrylamide [101]. Glycine and lysine can have a positive effect by competing with asparagine for the carbonyl group of the sugar moiety and/or forming adducts with acrylamide once it has formed. The SH group of cysteine (or other thiols) can benefit in two ways: forming an adduct with acrylamide and undergoing heat-induced H_2_S elimination to generate dehydroalanine [CH_2_=CH(NH_2_)COOH]. As with acrylamide, the NH_2_ group of asparagine can then engage in addition reactions with the dehydroalanine’s double bond in a competitive manner. In theory, serine can also be converted to dehydroalanine by removing H_2_O. The addition of antioxidants also reduces the acrylamide content in the food, influencing the Maillard reaction. Antioxidants extracted from bamboo leaves have been shown to decrease the acrylamide content, as they block the oxidation process of the Maillard reaction to a certain extent [102]. Enzymes such as asparaginase will significantly reduce the acrylamide content in the food as they hydrolyze the asparagine to aspartic acid without altering the taste of the food product, thereby reducing the concentration of the precursor necessary for the Maillard reaction [103]. As reported in a study, the salt solution can lower the acrylamide content in the food, such as NaCl and CaCl_2_, where the cations inhibit Schiff base formation by interacting with the free asparagine present in the food [104]. Acrylamide generation in the carbohydrate-rich food was reduced after adding vitamins due to their antioxidative properties. Vitamin C (ascorbic acid) and B_1_ (thiamine) reduced the acrylamide content in the food by almost 60%, and Vitamin B_2_ (riboflavin) and B_5_ (pantothenic acid) decreased approximately 30% of the acrylamide content when examined using the amino acid/sugar chemical model system [105]. Acrylamide content was decreased in food when the reducing sugars were replaced with the nonreducing sucrose powder; however, acrylamide content is reduced at very high temperatures due to the rate of degradation exceeding the rate of formation [13]. 

### 7.4. pH and Water Content

pH plays a vital role in controlling the acrylamide formation in the food product, as the Maillard reaction is mainly influenced by pH. Lowering the pH by adding acids like citric acid can enormously reduce acrylamide formation. Under acidic conditions, the non-protonated amine is converted to protonated amine, blocking the nucleophilic attack on the Schiff base and thereby reducing the possibility of the formation of acrylamide via the Maillard reactions [25]. The acrylamide formation decreased in the buffered solution of sodium acetate when unsaturated lipids were added to examine the effect of formation under acidic conditions [106]. The water content of the food product also impacts the formation of dietary acrylamide; the water activity is less than 0.4 of the acrylamide content in the food [1].

### 7.5. Fermentation

The fermentation process of the food product can limit the acrylamide content by altering the time taken for the fermentation. Some studies showed reduced acrylamide content in the food by 39% and 26% when processed under lactic acid fermentation using *Streptococcus lutetiensis* and *Lactobacillus plantarum*, respectively; it was also tested using in vitro digestion processes in the gastric stage, where it removed 30% of the acrylamide content, and in the intestine stage, it eliminated about 40%, as observed in a study by Albedwawi [107]. Moreover, another study reported that *Pediococcus acidilactici* lactic acid bacteria strain reduced the acrylamide content to 5.64 µg kg^−1^ in bread when inoculated and fermented for about 16 h. Lactic acid bacteria fermentation influenced the taste properties and increased the softness of the bread, as reported by Nachi et al. [108]. Since most people consume roasted coffee, the acrylamide content in the roasted coffee was controlled by using the yeast fermentation process, which reduced it by 70% when mixed with *Saccharomyces cerevisiae,* also known as baker’s yeast, and some sugar in a tightly closed container and fermented for 48 h at 30 °C, as reported by Akıllıoglu et al. [109]. Prolonged fermentation of carbohydrate-rich food products resulted in a decrease in acrylamide content, attributed to the increased utilization of asparagine over an extended duration.

### 7.6. Hydrocolloid-Based Coating

Hydrocolloids are long-chain polymers that are hydrophilic in nature, comprising amino and carbonyl groups with a higher molecular weight. Hydrocolloid coating reduces the acrylamide content of food products and is one of the natural and essential mitigation strategies for controlling the acrylamide content of food products [110]. Its ability to bind to lipids, oxygen, and carbon dioxide prevents excess oil absorption and decreases acrylamide formation during the frying process [98]. Hydrocolloid solutions showed positive results in reducing acrylamide formation, which increased the water retention capacity and hindered the Maillard reaction. It also increased the Margin of Exposure (MOE) of the coated French fries, which controlled the carcinogenic toxicity induced by dietary acrylamide [28]. In another study, the rate of acrylamide content decreased when it was coated with aqueous antioxidant plant extracts of *Zataria multiflora* and *Allium hirtifolium,* as they contain aldehyde groups in their structures, which limits the free asparagine. Along with the above method, the hydrocolloid coating using alginate and pectin also reduced the acrylamide content in the food as they lower the rate of heat transfer and core temperature of the food products, thereby preventing water evaporation and thus controlling the moisture content of the food, as reported by Zokaei et al. [111]. However, another study has also reported that a lower concentration of hydrocolloid coating reduced the level of acrylamide content in the fish nuggets in the presence of chitosan and gum Arabic, as they contain amino acids and proteins that would influence the formation of acrylamide in the fish crust [112].

### 7.7. Inhibitory and Inert Baking Atmosphere 

Using an inert and inhibitory baking atmosphere can reduce acrylamide formation. An anaerobic baking atmosphere using inert gases such as nitrogen and carbon dioxide decreased the acrylamide content by 50%, whereas inhibitory gases like sulfur dioxide (SO_2_) decreased the acrylamide content by 99%. It was observed that the reason for completely blocking the acrylamide formation in the bread when baked under an SO_2_ atmosphere is due to the sulfur atom’s nucleophilic structure, which binds more quickly to the carbonyl group of the reducing sugar than the amino acids, thereby hindering the Maillard reaction. Other than that, eliminating the oxygen from the baking atmosphere can also control the rate of the Maillard reaction. These atmospheric conditions also alter the sensory properties of the bread, where the sensorial properties of the sulfur dioxide atmosphere baked bread got worse compared to the inert gas atmosphere even though the rate of reduction of acrylamide content was higher, as reported by Gülcan et al. [113]. 

The formation of acrylamide content in the food can be controlled by utilizing various mitigation strategies, among which air and vacuum-frying are two of the most effective strategies with a reduction percentage of 72–98%, and the least effective is hydrocolloid coating, with a reduction percentage of 48%, as discussed above and shown in Table 2. Blanching or hydrocolloid coating followed by cooking at a controlled temperature and maintaining the acidic pH reduced the formation of acrylamide content in food. Moreover, utilizing additives and yeast fermentation is a very promising technology for controlling acrylamide formation by reducing the free asparagine molecules present in food products. Therefore, using mitigation strategies is one of the best ways to avoid the toxicity caused by the intake of acrylamide-rich food.

## 8. Conclusions

Acrylamide is one of the major toxic compounds formed in thermally processed food through the Maillard reaction. The high intake of acrylamide results in potential health risks. Extensive studies have explored factors contributing to the acrylamide in food. However, the majority of research has examined common variables like pH, moisture content, and temperature and is intended to concentrate more on the effects of specific/particular components that result in the browning of food. Hence, a comprehensive multi-technique strategy is essential for identifying and controlling these factors responsible for acrylamide formation. Several cutting-edge technologies, like hemoglobin nanoparticles, the E-tongue, the E-nose, etc., have been evaluated in laboratories but face challenges for large-scale industrial use. Therefore, advanced research should prioritize the development of simpler, cost-effective, highly sensitive, and quick detection methods for the identification of acrylamide in processed foods. As free asparagine and reducing sugars determine the formation of acrylamide, more research must be focused on altering the genes responsible for free asparagine and reducing sugars in the raw materials. Despite the widespread reliance on ready-to-eat, thermally processed meals for their convenience, flavor, and affordability, there is a need to raise awareness about the presence of thermally processed toxins in such foods. Furthermore, individuals should be informed about the potential health hazards associated with the intake of acrylamide-containing foods. Understanding the molecular mechanism and the factors involved in the generation of dietary acrylamide remains a significant challenge. 

## Figures and Tables

**Figure 1 foods-13-00556-f001:**
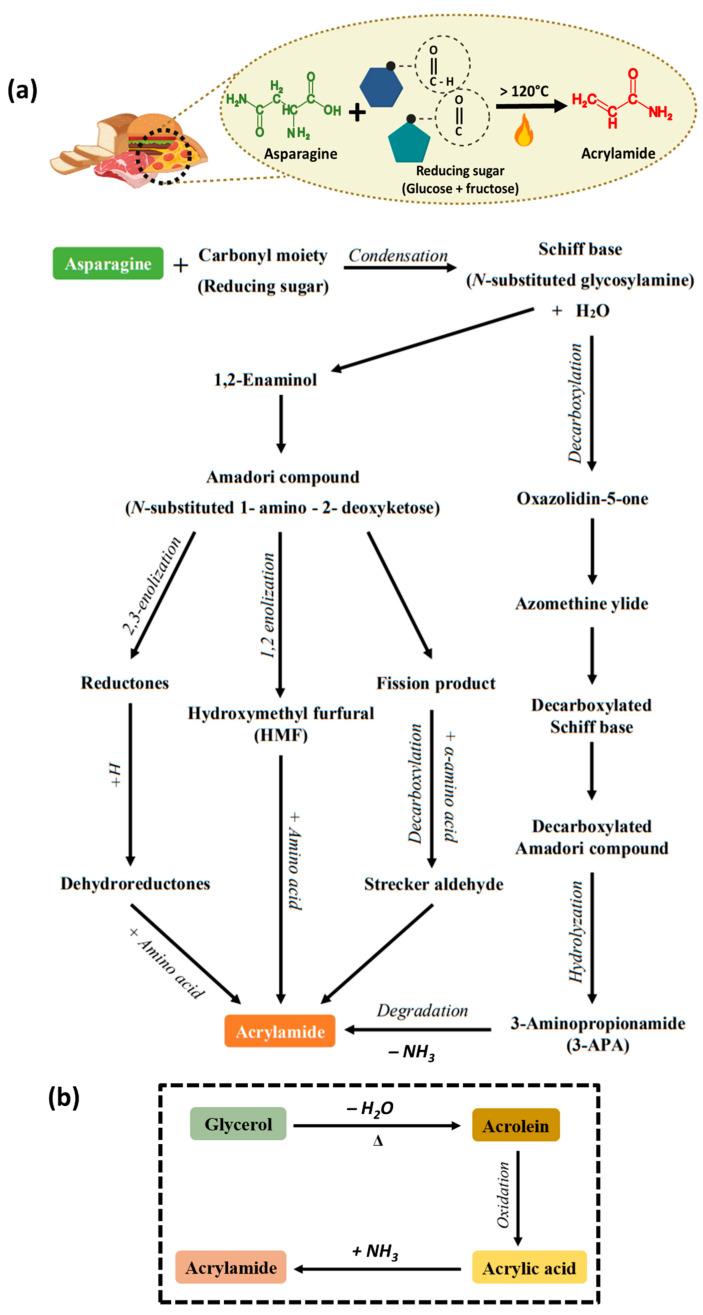
(**a**) Conversion of asparagine to acrylamide and the overall mechanism of acrylamide formation via the Maillard reaction. (**b**) Formation of acrylamide via the acrolein pathway.

**Figure 2 foods-13-00556-f002:**
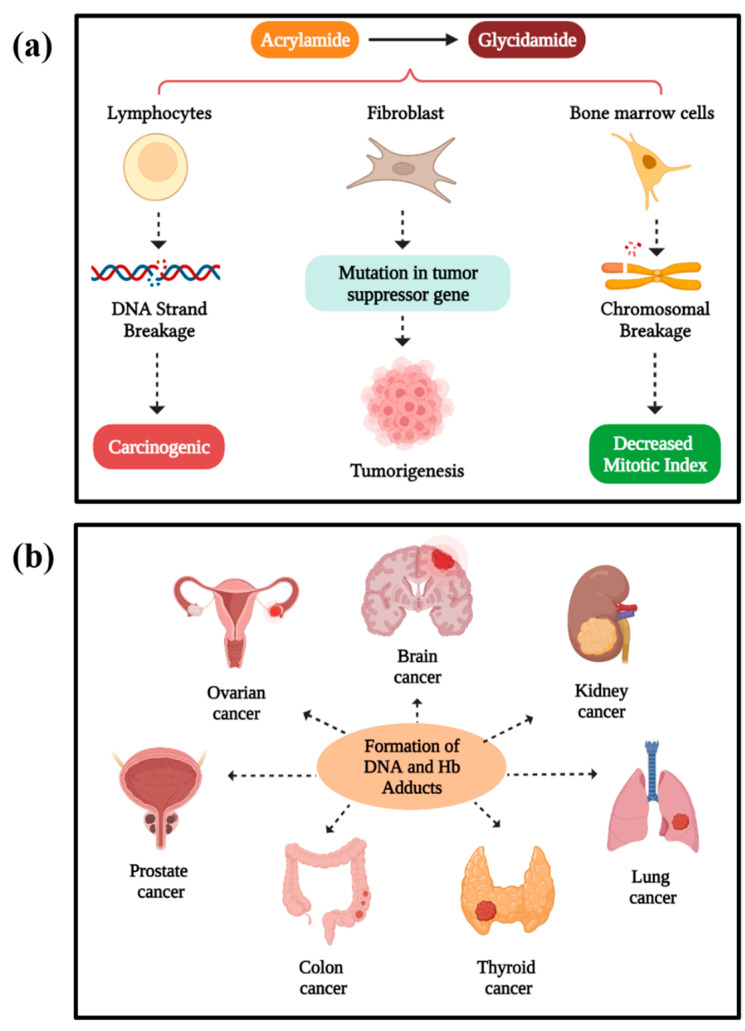
(**a**) Genotoxic effect of acrylamide and its metabolite glycidamide; (**b**) carcinogenic effect of acrylamide. Created with BioRender.com.

**Figure 3 foods-13-00556-f003:**
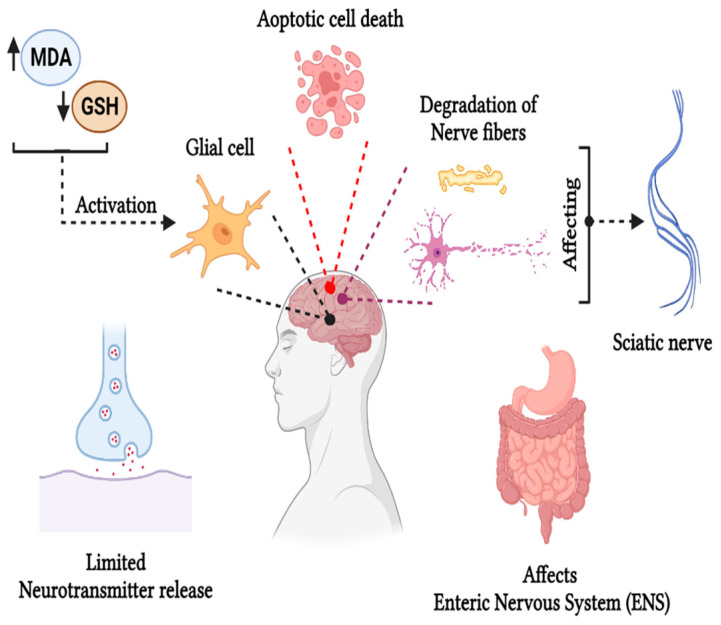
Neurotoxic effect of acrylamide created with BioRender.com.

**Figure 4 foods-13-00556-f004:**
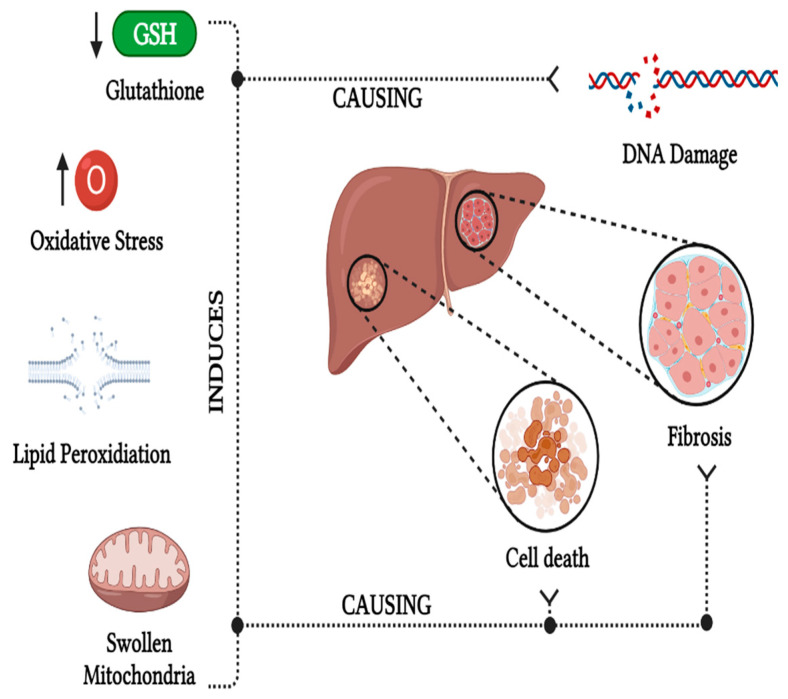
Hepatoxic effect of dietary acrylamide. Created with BioRender.com.

**Figure 5 foods-13-00556-f005:**
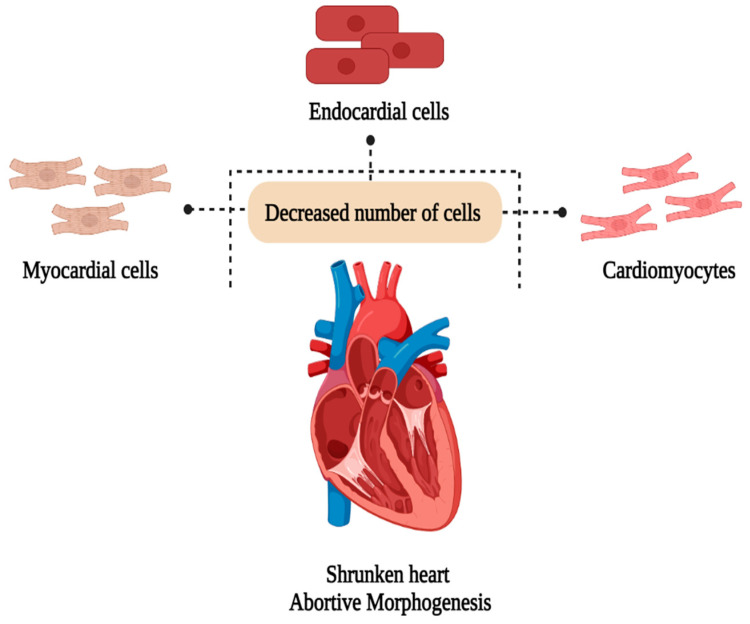
Cardiovascular toxicity of acrylamide. Created with BioRender.com.

**Figure 6 foods-13-00556-f006:**
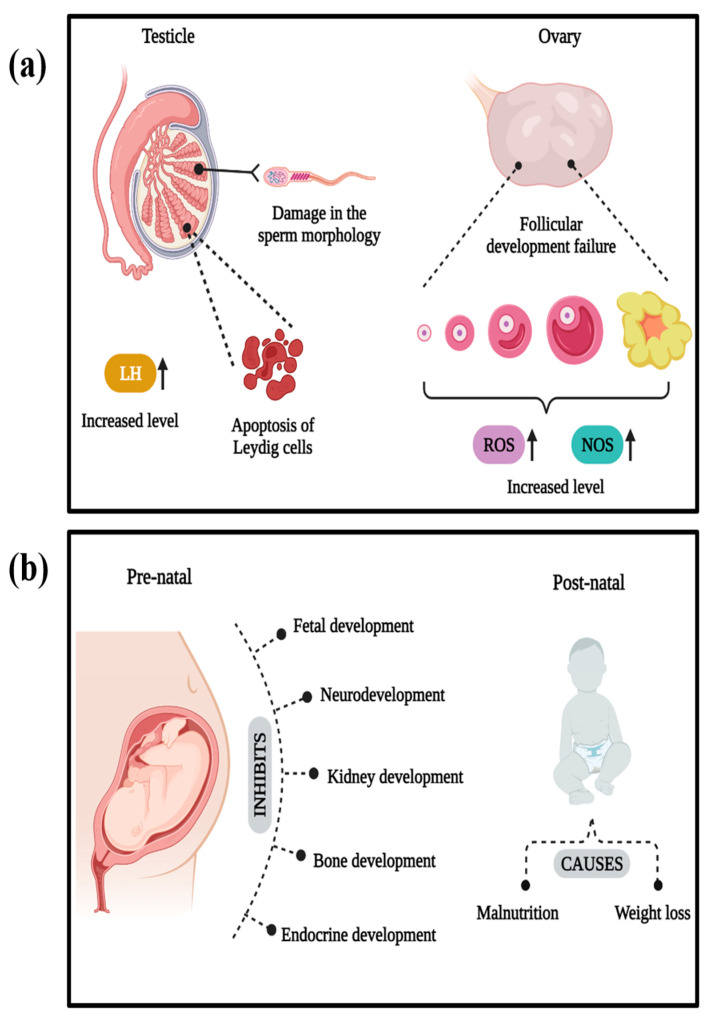
(**a**) Reproductive toxicity of acrylamide. (**b**) Prenatal and postnatal effects of acrylamide. Created with BioRender.com.

**Figure 7 foods-13-00556-f007:**
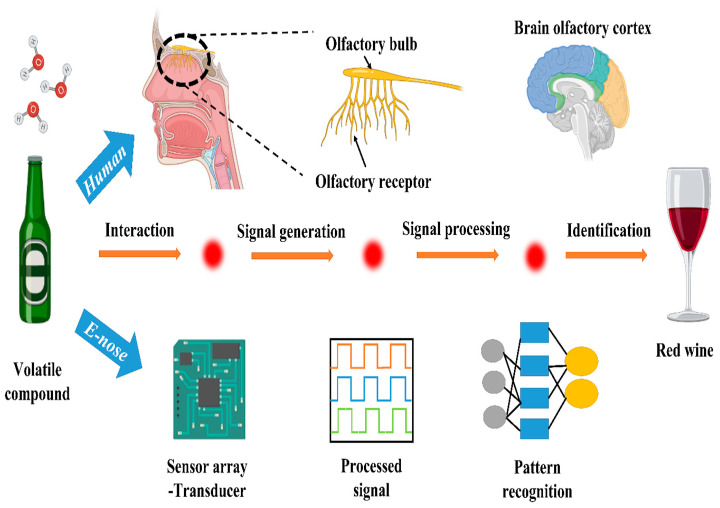
Schematic representation showing the working mechanism of E-nose for the detection of food materials.

**Table 1 foods-13-00556-t001:** Comparison data showing LOD and acrylamide detection range of various detection method.

Detection Method	Limit of Detection (LOD)	Range of Detection	Sample Model	References
E-tongue and nose	2.5 × 10^−3^ μg kg^−1^	2.5 × 10^−3^–20×10^−3^ μg kg^−1^	Olive and brine solution	[82]
Fluorescence biosensor	0.5 × 10^−6^ nmol L^−1^	0.05 M–10^−7^ nmol L^−1^	Potato fries	[94]
	2.41 × 10^−2^ nmol L^−1^	1 × 10^−1^–5 × 10^3^ nmol L^−1^	Bread crust	[95]
HbNPs	0.1 nmol L^−1^	0.05–100 nmol L^−1^	Bread, nuts, potato crips, biscuits	[91]
	0.06 nmol L^−1^	10–171 nmol L^−1^	French fries	[92]
SERS	2 µg kg^−1^	5–100 µg kg^−1^	Fried food	[96]
	0.02 nmol L^−1^	0.1–5 × 10^4^ nmol L^−1^	Cookies, chips, and bread	[97]
LC-MS/MS	3 ng mL^−1^	16.8–72.8 ng mL^−1^	Coffee	[84]
	6 µg kg^−1^	197–639 µg kg^−1^	Breakfast cereals	[85]
	2 μg kg^−1^	95.8–9826 μg kg^−1^	Medicine homologous foods (*Atractylodis Macrocephalae* Rhizoma)	[87]
GC-MS	0.6 µg kg^−1^	1–500 µg kg^−1^	Bread, biscuits, wafers, cakes, cookies, and crackers	[89]
	0.6 µg kg^−1^	33.36–250.90 µg kg^−1^	Roasted nuts and seeds	[90]

**Table 2 foods-13-00556-t002:** Comparison data showing the reduction percentage of acrylamide using various mitigation strategies.

Mitigation Strategies	Reduction Percentage (%)	Sample Model	Reference
Air- and vacuum-frying	72–98%	Potato chips	[100]
Blanching	65% and 96%	French fries and potato crisps	[27]
Additives	30–60%	Amino acid/sugar chemical model	[105]
Fermentation	70%	Roasted coffee	[109]
Hydrocolloid coating	48%	French fries	[28]
Inhibitory and inert baking atmosphere	50–99%	Bread	[113]

## Data Availability

Data is contained within the article.

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
