# Peer review of "Dietary Acrylamide: A Detailed Review on Formation, Detection, Mitigation, and Its Health Impacts"

_foods, 2024, doi:10.3390/foods13040556_

Round 1
Reviewer 1 Report
Comments and Suggestions for Authors
I congratulate the authors, the article is very good. One more thing can be added: in some countries there are legal measures regarding acrylamid ,(see a directive in EU). Maybe a small subchapter regarding these measures would be inspirational and informative.
Comments on the Quality of English Language
Please revise language, especially in abstract.
Author Response
Response to Reviewer 1
I congratulate the authors, the article is very good. One more thing can be added: in some countries there are legal measures regarding acrylamide ,(see a directive in EU). Maybe a small subchapter regarding these measures would be inspirational and informative.
Please revise language, especially in abstract.
Author’s Response: Thank you for your kind words and suggestions. We have included a small subsection of the legal measures regarding acrylamide in the introduction part of the revised manuscript. As there is no specific nationwide legal measures on acrylamide in food were in place in India, therefore it is not included. We have improved the language in the abstract in the revised manuscript.

Reviewer 2 Report
Comments and Suggestions for Authors
The authors present a review on the formation, detection, mitigation and the health impacts of the dietary acrylamide. The manuscript need major revision.
Specific comments:
1. Many reviews on acrylamide have already been published. The authors should emphasize the importance of the specific manuscript.
2. The language in not so smooth, and it needs polishing. The manuscript needs to be thoroughly proofread by an English language expert to eliminate several typos and grammatical errors.
3. Some of the sentences are too long, hence reduced clarity. The authors need to shorten the sentences to make the message clear.
4. Line 38. Molecular weight unit is gram per mole.
5. In the manuscript the expression “reported by” should by followed by the author’s name and not only the reference number.
6. Lines 231-235. This is a summary and should be mentioned in a separate paragraph.
7. EFSA’s opinion on the exposure of acrylamide should be included in the references.
8. Table 1. Check and correct the values for the range of detection. In some cases, the range of detection is lower than the LOD. The units should be the same for LOD and detection range.
9. Lines 631-634 are not relevant in the specific section and should be deleted.
10. Lines 743-753. This is a summary of the mitigation measures and should be in separate paragraph.
11. Lines 614-618 should be removed from section 6.7.
12. Delete lines 492-496 from section 5. The detection techniques are discussed in section 6.
13. References should be checked regarding the format.
14. Rephrase “consuming dietary acrylamide”. Better use acrylamide intake rather than acrylamide consumption. Correct throughout the manuscript.
Comments on the Quality of English Language
The language in not so smooth, and it needs polishing. The manuscript needs to be thoroughly proofread by an English language expert to eliminate several typos and grammatical errors.
Author Response
Response to Reviewer 2
The authors present a review on the formation, detection, mitigation and the health impacts of the dietary acrylamide. The manuscript needs major revision.
Specific comments:
- Many reviews on acrylamide have already been published. The authors should emphasize the importance of the specific manuscript.
Author’s Response: We thank the reviewer for the suggestion. We have modified the manuscript as suggested and importance of the review is explained in the revised manuscript.
- The language in not so smooth, and it needs polishing.The manuscript needs to be thoroughly proofread by an English language expert to eliminate several typos and grammatical errors.
Author’s Response: Thank you for the suggestion. We have corrected spelling and grammatical errors as pointed out by the reviewer. The revised manuscript now reflects these improvements for a smoother language experience.
- Some of the sentences are too long, hence reduced clarity. The authors need to shorten the sentences to make the message clear.
Author’s Response: Thank you for your feedback. We have addressed the concern regarding sentence length to improve clarity, and the revised manuscript now features shorter sentences for enhanced readability.
- Line 38. Molecular weight unit is gram per mole.
Author’s Response: Thank you for pointing out the clarification regarding the molecular weight unit. We have updated the unit of molecular weight in revised manuscript.
- In the manuscript the expression “reported by” should by followed by the author’s name and not only the reference number.
Author’s Response: Thank you for your valuable feedback. We have revised the manuscript to ensure that when using the expression "reported by," it is now followed by the author's name rather than just the reference number.
- Lines 231-235. This is a summary and should be mentioned in a separate paragraph.
Author’s response: Thank you for highlighting the need for a separate paragraph for the summary. We have incorporated this suggestion.
- EFSA’s opinion on the exposure of acrylamide should be included in the references.
- Table 1. Check and correct the values for the range of detection. In some cases, the range of detection is lower than the LOD. The units should be the same for LOD and detection range.
Author’s response: We thank the reviewer for the suggestion. We have updated the table 1 in the revised manuscript.
- Lines 631-634 are not relevant in the specific section and should be deleted.
Author’s response: Thank you for bringing this to our attention. We have reviewed lines 631-634 and agree that they are not relevant to the specific section. Accordingly, we have deleted them.
- Lines 743-753. This is a summary of the mitigation measures and should be in separate paragraph.
Author’s response: Thank you for highlighting the need for a separate paragraph for the summary. We have incorporated this suggestion.
- Lines 614-618 should be removed from section 6.7.
Author’s Response: We have carefully reviewed the manuscript and removed lines 614-618
- Delete lines 492-496 from section 5. The detection techniques are discussed in section 6.
Author’s Response: As suggested, the mentioned lines are deleted.
- References should be checked regarding the format.
Author’s Response: Thank you for your suggestion. We have checked the format of the references and modified accordingly.
- Rephrase “consuming dietary acrylamide”. Better use acrylamide intake rather than acrylamide consumption. Correct throughout the manuscript.
Author’s Response: Thank you for your suggestion. We have replaced "consuming dietary acrylamide" with "acrylamide intake" throughout the revised manuscript.
- Comments on the Quality of English Language
The language in not so smooth, and it needs polishing. The manuscript needs to be thoroughly proofread by an English language expert to eliminate several typos and grammatical errors.
Author’s Response: Thank you for highlighting the need for language refinement in our manuscript. We appreciate your valuable suggestion and have taken the necessary steps to thoroughly proofread the document, addressing typos and grammatical errors. The manuscript has undergone scrutiny. We have tracked the changes made.

Reviewer 3 Report
Comments and Suggestions for Authors
Comments to the Authors
The problem of toxic/carcinogenic organic compounds, including acrylamide (AA) which occur in the diet, and how to protect their synthesis in food, belongs to current scientific issues which are vital for human health safety.
Below you will find the list of remarks which I propose to take into consideration while preparing a revised version of the manuscript.
General remark: In the descriptions of all drawings, please indicate the references on which they are based. And also, if there are abbreviations in the drawings, please explain them.
Please standardize the units of concentration throughout the paper, such as 0.1 nM or 0.1 nmol L-1 or 0.1 nmol/L. Similarly: ng/g or ng g-1, and ug/kg or ng/g, etc.
Introduction
Line 50-51: “Dietary carbohydrates can control human metabolism by decreasing sugars, which are the primary source of energy” I don't understand this sentence. Carbohydrates are sugars. Please complete or correct this sentence.
Line 62: Please complete: “concentration” of what?
Line 70-72: The sentence “In addition… t in determining dietary acrylamide in food and other sources” The sentence refers to one way of detecting acrylamide. It was placed between information on formation in food and exposure from other sources. Please check that you have not omitted a portion of the text.
3. Mechanism of acrylamide formation in food
In Figure C above, the arrow should probably be - H20, instead of - H2?
The point (a) shown in Figure 1 is detailed in point (b). I propose to clarify this or combine (a) and (b), because the current version of the figure suggests that there are 3 different patterns of AA formation.
Line 100: The sentence “The second most typical pathway for acrylamide formation in food is
called the acrolein pathway” can be moved to line 123, where this mechanism of AA formation is discussed in more detail. Please also indicate in the text the number of the figure showing this mechanism.
Line 132: It would be more correct to write that glycerol is “dehydrated”, instead “degraded” to acrolein. (Please see for example: https://doi.org/10.1021/jp060597q)
Line 138 – 149: I propose moving the text on the occurrence of AA in food and the factors affecting its formation to subsection 4.
4. Factors influencing the formation of AA
Line 165: Please explain the mechanism of “Micheal addition,” or omit this sentence.
Line 167-169: this sentence is about the impact of sugar, not temperature and time. Please move to the appropriate place in the text
4.2. pH and surface area: It is worth creating separate subsections on pH and the second on surface area
5. Diseases related to acrylamide exposure
Line 241. Provide original link instead of [9]
6. Detection of acrylamide in food
General Note: I propose that this chapter focus on methods, their advantages, disadvantages, limits of quantification, detection range (as in Table 1), omitting details of AA concentrations in food or moving this information to Chapter 7.
Figure 7 applies only to E-nose. Please correct.
Line 503-504: HPLC is a liquid chromatography technique, as is LC, which is discussed in a separate chapter (6.3). It is a technique for separating components of mixtures, not a detection method. When liquid chromatography is combined with mass spectrometry MS or UV (e.g., with a diode array DAD or fluorescence detector), it becomes a technique for determining analytes.
I propose the creation of one subsection, with the title, for example, Liquid Chromatography in combination with various detection methods.
Line 546: “HPLC-UV is another way of determination of acrylamide in food with high rate of extraction speed”
I would like to point out that when using DAD we are also measuring the absorbance of UV light, so HPLC-UV is not an “another way” of determination. HPLC - DAD and HPLC-UV involve the determination of compounds at selected UV wavelengths. These are the same methods of determination. Please correct the text.
Line 542 – “HPLC, when combined with 7.3 mM 2-naphthalenethiol, which acts as a derivatization reagent,…”
I would like to note that HPLC is a separation technique, and 2-naphthalenethiol is a derivatization reagent. It is not possible to combine a reagent with a technique. Please correct this passage. Besides, it doesn't really matter in this review what the concentration of 2-naphthalenethiol used for AA derivatization is, but it would be more interesting to know at what wavelength this acrylamide derivative was determined, or how the sensitivity of the determination method has changed (improved).
Line 545: “HPLC-UV is another way of determination of acrylamide in food with high rate of extraction speed”
I would like to note that HPLC does not affect the extraction rate. Determinations by the HPLC technique are performed for compounds present in extracts previously isolated from the sample.
Line 549: It is worth mentioning that LC-MS is one of the most precise and accurate methods
Line 589: please standardize the LOD units in this chapter
7. Mitigation strategies of dietary acrylamide
Line 662-664: Explain in more detail what is the mechanism of inhibition of AA synthesis by the listed amino acids:
“The addition of amino acids such as lysine, glycine, and alanine reduced the acrylamide content in the food as it has a nucleophilic component that binds covalently, thereby eliminating the acrylamide as reported by [107]”
Line 686: please check whether it should be “the nucleophilic attack” instead of "neutrophilic”
Author Response
Response to Reviewer 3
Comments to the Authors
The problem of toxic/carcinogenic organic compounds, including acrylamide (AA) which occur in the diet, and how to protect their synthesis in food, belongs to current scientific issues which are vital for human health safety.
Below you will find the list of remarks which I propose to take into consideration while preparing a revised version of the manuscript.
General remark: In the descriptions of all drawings, please indicate the references on which they are based. And also, if there are abbreviations in the drawings, please explain them.
Please standardize the units of concentration throughout the paper, such as 0.1 nM or 0.1 nmol L-1 or 0.1 nmol/L. Similarly: ng/g or ng g-1, and ug/kg or ng/g, etc.
Authors’ response: We thank the reviewer for the suggestion. We have standardized the majority of the units of concentration as suggested in the revised manuscript.
Introduction
Line 50-51: “Dietary carbohydrates can control human metabolism by decreasing sugars, which are the primary source of energy” I don't understand this sentence. Carbohydrates are sugars. Please complete or correct this sentence.
Author’s Response: Thank you for bringing this to our attention. We have updated the sentences in the revised manuscript.
Line 62: Please complete: “concentration” of what?
Author’s Response: Thank you for your feedback. Here, ‘concentration’ refers to concentration of the starchy/carbohydrate foods.
Line 70-72: The sentence “In addition… t in determining dietary acrylamide in food and other sources” The sentence refers to one way of detecting acrylamide. It was placed between information on formation in food and exposure from other sources. Please check that you have not omitted a portion of the text.
Author’s Response: Thank you for your feedback, we have placed it in the relevant section.
- Mechanism of acrylamide formation in food
In Figure C above, the arrow should probably be - H20, instead of - H2?
Author’s Response: Thank you for your comment, we have revised it.
The point (a) shown in Figure 1 is detailed in point (b). I propose to clarify this or combine (a) and (b), because the current version of the figure suggests that there are 3 different patterns of AA formation.
Author’s Response: We thank the reviewer for the comment. Fig 1 (a) & (b) is combined as suggested by the reviewer. However, fig (c) represents another minor pathway of the acrylamide formation through the formation of acrolein and hence it is kept separately.
Line 100: The sentence “The second most typical pathway for acrylamide formation in food is called the acrolein pathway” can be moved to line 123, where this mechanism of AA formation is discussed in more detail. Please also indicate in the text the number of the figure showing this mechanism.
Author’s Response: Thank you for your comment, we have revised the sentence.
Line 132: It would be more correct to write that glycerol is “dehydrated”, instead “degraded” to acrolein. (Please see for example: https://doi.org/10.1021/jp060597q)
Author’s Response: Thank you for your suggestion, and we have incorporated the above suggestion.
Line 138 – 149: I propose moving the text on the occurrence of AA in food and the factors affecting its formation to subsection 4.
Author’s Response: Thank you for the input, we have omitted out the lines from 138-149 in the revised manuscript.
- Factors influencing the formation of AA
Line 165: Please explain the mechanism of “Micheal addition,” or omit this sentence.
Author’s Response: Thank you for your suggestion, we have omitted the corresponding sentence.
Line 167-169: this sentence is about the impact of sugar, not temperature and time. Please move to the appropriate place in the text 4.2. pH and surface area: It is worth creating separate subsections on pH and the second on surface area
Author’s Response: Thank you, we have made the necessary changes in the revised manuscript.
- Diseases related to acrylamide exposure
Line 241. Provide original link instead of [9]
Author’s Response: Thank you, we have made the necessary changes. https://apps.who.int/food-additives-contaminants-jecfa-database/Home/Chemical/5198
https://www.iarc.who.int/branches-nme-research/
- Detection of acrylamide in food
General Note: I propose that this chapter focuses on methods, their advantages, disadvantages, limits of quantification, detection range (as in Table 1), omitting details of AA concentrations in food or moving this information to Chapter 7.
Author’s Response: Thank you for the comment. Section 6 deals with the detection strategies for acrylamide and hence we feel table-1 fits in the section-6.
Figure 7 applies only to E-nose. Please correct.
Author’s Response: Thank you for the correction, we have made changes in the caption.
Line 503-504: HPLC is a liquid chromatography technique, as is LC, which is discussed in a separate chapter (6.3). It is a technique for separating components of mixtures, not a detection method. When liquid chromatography is combined with mass spectrometry MS or UV (e.g., with a diode array DAD or fluorescence detector), it becomes a technique for determining analytes.
I propose the creation of one subsection, with the title, for example, Liquid Chromatography in combination with various detection methods.
Author’s Response: Thank you for your valuable input, we have changed the title as mentioned above.
Line 546: “HPLC-UV is another way of determination of acrylamide in food with high rate of extraction speed”
I would like to point out that when using DAD we are also measuring the absorbance of UV light, so HPLC-UV is not an “another way” of determination. HPLC - DAD and HPLC-UV involve the determination of compounds at selected UV wavelengths. These are the same methods of determination. Please correct the text.
Author’s Response: Thank you for the correction, we have revised as mentioned.
Line 542 – “HPLC, when combined with 7.3 mM 2-naphthalenethiol, which acts as a derivatization reagent,…”
I would like to note that HPLC is a separation technique, and 2-naphthalenethiol is a derivatization reagent. It is not possible to combine a reagent with a technique. Please correct this passage. Besides, it doesn't really matter in this review what the concentration of 2-naphthalenethiol used for AA derivatization is, but it would be more interesting to know at what wavelength this acrylamide derivative was determined, or how the sensitivity of the determination method has changed (improved).
Line 545: “HPLC-UV is another way of determination of acrylamide in food with high rate of extraction speed” I would like to note that HPLC does not affect the extraction rate. Determinations by the HPLC technique are performed for compounds present in extracts previously isolated from the sample.
Author’s Response: Thank you for the comment. We have completely removed the HPLC section. We agree with the reviewer, more than detection, it is a method for separating components of mixtures.
Line 549: It is worth mentioning that LC-MS is one of the most precise and accurate methods.
Author’s Response: Thank you for the comment. We have included as reviewer’s suggestion.
Line 589: please standardize the LOD units in this chapter
Author’s Response: Thank you for the comment. We have modified it as reviewer’s suggestion.
- Mitigation strategies of dietary acrylamide
Line 662-664: Explain in more detail what is the mechanism of inhibition of AA synthesis by the listed amino acids “The addition of amino acids such as lysine, glycine, and alanine reduced the acrylamide content in the food as it has a nucleophilic component that binds covalently, thereby eliminating the acrylamide as reported by [107]”
Author’s Response: We thank the reviewer for their comment. We have added the mechanism of inhibition as suggested.
One or two acrylamide molecules can alkylate the ε-NH2 group of lysine to form mono- and disubstituted lysine derivatives, respectively. The monosubstituted protein-bound lysine derivative would form Nε-(2-carbamoylethyl) lysine on enzymatic hydrolysis and Nε-(2-carboxylethyl) lysine [HOOCCH2CH2NH(CH2)4CH(NH2)COOH] on acid hydrolysis.
Also, Glycine and lysine can have a positive effect by competing with asparagine for the carbonyl group of the sugar moiety and/or forming adducts with acrylamide once it has formed. The SH group of cysteine (or other thiols) can benefit in two ways: forming an adduct with acrylamide and undergoing heat-induced H2S elimination to generate dehydroalanine [CH2═CH(NH2)COOH]. As with acrylamide, the NH2 group of asparagine can then engage in addition reactions with the dehydroalanine's double bond in a competitive manner. In theory, serine can also be converted to dehydroalanine by removing H2O.
Line 686: please check whether it should be “the nucleophilic attack” instead of "neutrophilic.”
Author’s Response: Thank you for the correction, we have corrected the error in the revised manuscript.

Reviewer 4 Report
Comments and Suggestions for Authors
This review paper would benefit from greater focus. While the authors attempt to provide a comprehensive overview of acrylamide research, the current presentation is overly general.
Line 12: Consider removing the word "adulterated" or providing a supporting reference for this assertion.
Line 59: Clarify the study's endpoint, such as the level of acrylamide produced or the effect of the treatment on acrylamide formation.
Line 77: Disagree with this statement. Instead, summarize the previous revisions and provide a clear explanation of the novel findings presented in the manuscript. ( see: 10.1177/109158182090240; doi.org/10.1080/87559129.2020.1719505 and others)
The introduction is excessively lengthy. Condense it to offer a clear overview of the review's main topics without delving into excessive detail.
Line 93_ Acrylamide Formation: Provide a more in-depth explanation of the three pathways for acrylamide formation, including the specific enzymes and precursors involved.
Line 153_ Factors Affecting Acrylamide Formation: Include a section on the impact of food matrix and composition on acrylamide formation.
Line 166- What is lower temperature? Section 4: "Effect of Processing Conditions": Provide more specific examples of how processing conditions, such as temperature, time, and moisture content, influence acrylamide formation.
Section 5: "Analytical Methods": Elaborate on the various methods used to analyze acrylamide in food samples, including their strengths, weaknesses, and applications.
Section 6: "Extraction and Sample Preparation Techniques": Supply detailed descriptions of the extraction and sample preparation techniques employed for liquid and solid samples, including their limitations and potential biases.
Section 7: "Mitigation Strategies": Expand on the mitigation strategies for reducing acrylamide formation in food products, encompassing cooking methods, ingredient selection, and technological interventions.
Author Response
Response to Reviewer 4
This review paper would benefit from greater focus. While the authors attempt to provide a comprehensive overview of acrylamide research, the current presentation is overly general.
Line 12: Consider removing the word "adulterated" or providing a supporting reference for this assertion.
Author's Response: Thank you for the comment, we have removed the word adulterated from line 12 in the revised manuscript.
Line 59: Clarify the study's endpoint, such as the level of acrylamide produced or the effect of the treatment on acrylamide formation.
Author’s Response: We thank the reviewer for the comment. We have discussed this in section 6 in detail under ‘detection of acrylamide’. Levels of acrylamide produced are marked in the blue for your reference.
Line 77: Disagree with this statement. Instead, summarize the previous revisions and provide a clear explanation of the novel findings presented in the manuscript. ( see: 10.1177/109158182090240; doi.org/10.1080/87559129.2020.1719505 and others)
Author’s Response: We thank the reviewer for the comment. We have modified it accordingly.
The introduction is excessively lengthy. Condense it to offer a clear overview of the review's main topics without delving into excessive detail.
Author's Response: We thank the reviewer for the comment. We have modified it accordingly.
Line 93_ Acrylamide Formation: Provide a more in-depth explanation of the three pathways for acrylamide formation, including the specific enzymes and precursors involved.
Author’s Response: We thank the reviewer for the comment. We have discussed the pathways with figures as suggested.
Line 153_ Factors Affecting Acrylamide Formation: Include a section on the impact of food matrix and composition on acrylamide formation.
Author’s Response: We have discussed the impact of food composition on the acrylamide formation in the relevant section as suggested.
Line 166- What is lower temperature? Section 4: "Effect of Processing Conditions": Provide more specific examples of how processing conditions, such as temperature, time, and moisture content, influence acrylamide formation.
Author's Response: We thank the reviewer for the suggestion. We have modified them in the revised manuscript.
Section 5: "Analytical Methods": Elaborate on the various methods used to analyze acrylamide in food samples, including their strengths, weaknesses, and applications.
Author's Response: We thank the reviewer for the suggestion. We have modified them in the revised manuscript.
Section 6: "Extraction and Sample Preparation Techniques": Supply detailed descriptions of the extraction and sample preparation techniques employed for liquid and solid samples, including their limitations and potential biases.
Author's Response: We thank the reviewer for the suggestion. We have modified them in detail the revised manuscript.
Section 7: "Mitigation Strategies": Expand on the mitigation strategies for reducing acrylamide formation in food products, encompassing cooking methods, ingredient selection, and technological interventions.
Author's Response: We thank the reviewer for the suggestion. We have modified them in detail the revised manuscript.

Round 2
Reviewer 2 Report
Comments and Suggestions for Authors
The authors revised the manuscript according to reviewers’ comments. Most of the reviewers’ comments/suggestions have been answered and the corrections and additions are included in the revised manuscript. The manuscript is required minor revision.
Specific comments:
1. In the manuscript the expression “reported by” should by followed by the author’s name and not only the reference number.
2. EFSA’s opinion on the exposure of acrylamide should be included in the references.
3. Rephrase “consuming dietary acrylamide”. Better use acrylamide intake rather than acrylamide consumption. Correct throughout the manuscript.
Author Response
Response to Reviewer 2
The authors revised the manuscript according to reviewers’ comments. Most of the reviewers’ comments/suggestions have been answered and the corrections and additions are included in the revised manuscript. The manuscript is required minor revision.
Specific comments:
- In the manuscript the expression “reported by” should by followed by the author’s name and not only the reference number.
Author’s response: Thank you for your suggestion. We have done the above correction in the revised manuscript.
- EFSA’s opinion on acrylamide exposure should be included in the references.
Author’s response: Thank you for the comment, we have included the mentioned reference at the end of the revised manuscript.
- Rephrase “consuming dietary acrylamide”. Better use acrylamide intake rather than acrylamide consumption. Correct throughout the manuscript.
Author’s response: Thank you for your comment. We have replaced “consuming dietary acrylamide” with acrylamide intake throughout the revised manuscript.

Reviewer 3 Report
Comments and Suggestions for Authors
Figure 1
The authors corrected figure 1, but the description under the figure has not been changed. the caption still lists 3 pathways (a, b, c), although in the corrected figure there are only a and b. Please correct the caption.
Line 542 - Still the Maillard reaction mechanism (Fig 1a) of AA synthesis is given in the text as Fig 1(b). Correct please.
Line 1975 - Do not list HPLC among the detection methods. The corrected version still contains this error. HPLC is a technique for separating components of mixtures. Please just remove HPLC from this sentence.
Table 1
In the table 1, units are still expressed in different units. Decide on one way to specify units please.. E.g.
0.5 × 10−6 M = 0.5 × 10−6 mol/L = 0.5 × 10−6 mol L-1
Or
0.06 nmol L-1 = 0.06 nM = 0.06 nmol/L
Or
2 µg/kg = 2 µg kg-1 = 2 ng/g = 2 ng g-1
0.6 ng g−1 = 0.6 ng/g = 0.6 µg/kg =0.06 µg kg-1
Please also decide whether the authors want to present units in mol/l or nmol/l, similarly whether in ng/g or ug/kg
This also applies to the rest of the text. for example, in the line 2305 is 0.02 nM and in the line 2312 is 0.06 nmol L-1
Author Response
Response to Reviewer 3
Figure 1
The authors corrected figure 1, but the description under the figure has not been changed. the caption still lists 3 pathways (a, b, c), although in the corrected figure there are only a and b. Please correct the caption.
Author’s response: Thank you for your correction, we have made the above-mentioned changes in the revised manuscript.
Line 542 - Still the Maillard reaction mechanism (Fig 1a) of AA synthesis is given in the text as Fig 1(b). Correct please.
Author’s response: Thank you for your feedback, we have corrected the mistake in the revised manuscript.
Line 1975 - Do not list HPLC among the detection methods. The corrected version still contains this error. HPLC is a technique for separating components of mixtures. Please just remove HPLC from this sentence.
Author’s response: Thank you for the comment, we have removed “HPLC” from the mentioned sentence.
Table 1
In the table 1, units are still expressed in different units. Decide on one way to specify units please.. E.g.
0.5 × 10−6 M = 0.5 × 10−6 mol/L = 0.5 × 10−6 mol L-1
Or
0.06 nmol L-1 = 0.06 nM = 0.06 nmol/L
Or
2 µg/kg = 2 µg kg-1 = 2 ng/g = 2 ng g-1
0.6 ng g−1 = 0.6 ng/g = 0.6 µg/kg =0.06 µg kg-1
Please also decide whether the authors want to present units in mol/l or nmol/l, similarly whether in ng/g or ug/kg
This also applies to the rest of the text. for example, in the line 2305 is 0.02 nM and in the line 2312 is 0.06 nmol L-1
Author’s response: Thank you for your feedback. We have made the above-mentioned units in the revised manuscript.
|
Detection method |
Limit of detection (LOD) |
Range of detection |
Sample model |
References |
|
E-tongue and nose |
2.5×10-3 μg kg-1 |
2.5×10-3 -20×10-3 μg kg -1 |
Olive and brine solution |
[85] |
|
Fluorescence biosensor |
0.5 × 10−6 nmol L-1 |
0.05 M - 10−7 nmol L-1 |
Potato fries |
[100] |
|
2.41 × 10− 2 nmol L-1 |
1 × 10− 1 - 5 × 103 nmol L-1 |
Bread crust |
[101] |
|
|
HbNPs |
0.1 nmol L-1 |
0.05 - 100 nmol L-1 |
Bread, nuts, potato crips, biscuits |
[97] |
|
0.06 nmol L-1 |
10 - 171 nmol L-1 |
French fries |
[98] |
|
|
SERS |
2 µg kg−1 |
5–100 µg kg−1 |
Fried food |
[102] |
|
0.02 nmol L-1 |
0.1 - 5×104 nmol L-1 |
Cookies, chips, and bread |
[103] |
|
|
LC-MS/MS |
3 ng mL−1 |
16.8–72.8 ng mL−1 |
Coffee |
[91] |
|
6 µg kg-1 |
197 - 639 µg kg-1 |
Breakfast cereals |
[92] |
|
|
2 μg kg-1 |
95.8 - 9826 μg kg-1 |
Medicine homologous foods (Atractylodis Macrocephalae Rhizoma) |
[94] |
|
|
GC-MS |
0.6 µg kg-1 |
1–500 µg kg-1 |
Bread, biscuits, wafers, cakes, cookies, and crackers |
[95] |
|
0.6 µg kg-1 |
33.36 - 250.90 µg kg-1 |
Roasted nuts and seeds |
[96] |

Reviewer 4 Report
Comments and Suggestions for Authors
The revision focused mainly on English language improvements, and unfortunately, few of the previous comments were considered. The manuscript still has significant shortcomings.
Comments on the Quality of English Language
No comments
Author Response
Response to Reviewer 4
The revision focused mainly on English language improvements, and unfortunately, few of the previous comments were considered. The manuscript still has significant shortcomings.
Author's Response: We thank the reviewer for the suggestion. We have made the necessary corrections to the revised manuscript.